# (Re-)Emergence of Oropouche Virus (OROV) Infections: Systematic Review and Meta-Analysis of Observational Studies

**DOI:** 10.3390/v16091498

**Published:** 2024-09-22

**Authors:** Matteo Riccò, Silvia Corrado, Marco Bottazzoli, Federico Marchesi, Renata Gili, Francesco Paolo Bianchi, Emanuela Maria Frisicale, Stefano Guicciardi, Daniel Fiacchini, Silvio Tafuri, Antonio Cascio, Pasquale Gianluca Giuri, Roberta Siliquini

**Affiliations:** 1AUSL–IRCCS di Reggio Emilia, Servizio di Prevenzione e Sicurezza Negli Ambienti di Lavoro (SPSAL), Local Health Unit of Reggio Emilia, 42122 Reggio Emilia, Italy; 2ASST Rhodense, Dipartimento della Donna e Area Materno-Infantile, UOC Pediatria, 20024 Milan, Italy; scorrado@asst-rhodense.it; 3Department of Otorhinolaryngology, APSS Trento, 38122 Trento, Italy; marco.bottazzoli@apss.tn.it; 4Department of Medicine and Surgery, University of Parma, 43126 Parma, Italy; federico.marchesi@unipr.it; 5Department of Prevention, Turin Local Health Authority, 10125 Turin, Italy; renata.gili@aslcittaditorino.it; 6Health Prevention Department, Local Health Authority of Brindisi, 72100 Brindisi, Italy; frapabi@gmail.com; 7Ex Directorate General of Health Prevention, Ministry of Health, 00144 Rome, Italy; em.frisicale@sanita.it; 8Health Directorate, Local Health Authority of Bologna, 40124 Bologna, Italy; 9AST Ancona, Prevention Department, UOC Sorveglianza e Prevenzione Malattie Infettive e Cronico Degenerative, 61100 Ancona, Italy; daniel.fiacchini@sanita.marche.it; 10Department of Interdisciplinary Medicine, Aldo Moro University of Bari, 70121 Bari, Italy; silvio.tafuri@uniba.it; 11Infectious and Tropical Diseases Unit, Department of Health Promotion, Mother and Child Care, Internal Medicine and Medical Specialties, G D’Alessandro, University of Palermo, AOUP P. Giaccone, 90127 Palermo, Italy; antonio.cascio03@unipa.it; 12Department of Medicine and Diagnostics, AUSL di Parma, 43100 Parma, Italy; pgiuri@ausl.pr.it; 13Department of Public Health and Pediatric Sciences, University of Turin, 10126 Turin, Italy; 14Azienda Ospedaliera Universitaria City of Health and Science of Turin, 10126 Turin, Italy

**Keywords:** Oropouche Virus, Orthobunyavirus, arbovirus, vector-borne disease

## Abstract

Oropouche Virus (OROV; genus of Orthobunyavirus) is the causal agent of Oropouche Fever (OF). Due to the lack of specific signs and symptoms and the limited availability of diagnostic tests, the actual epidemiology of OROV infections and OF has been extensively disputed. In this systematic review with meta-analysis, a literature search was carried out in PubMed, Scopus, EMBASE, and MedRxiv in order to retrieve relevant articles on the documented occurrence of OROV infections. Pooled detection rates were then calculated for anti-OROV antibodies and virus detection (i.e., viral RNA detected by viral cultures and/or real-time polymerase chain reaction [RT-qPCR]). Where available, detection rates for other arboviruses (i.e., Dengue [DENV], Chikungunya [CHKV], and Zika Virus [ZIKV]) were calculated and compared to those for OROV. A total of 47 studies from South America and the Caribbean were retrieved. In individuals affected by febrile illness during OROV outbreaks, a documented prevalence of 0.45% (95% confidence interval [95%CI] 0.16 to 1.12) for virus isolation, 12.21% (95%CI 4.96 to 27.09) for seroprevalence (including both IgM and IgG class antibodies), and 12.45% (95%CI 3.28 to 37.39) for the detection of OROV-targeting IgM class antibodies were eventually documented. In the general population, seroprevalence was estimated to be 24.45% (95%CI 7.83 to 55.21) for IgG class antibodies. The OROV detection rate from the cerebrospinal fluids of suspected cases of viral encephalitis was estimated to be 2.40% (95%CI 1.17 to 5.03). The occurrence of OROV infections was consistently lower than that of DENV, CHKV, and ZIKV during outbreaks (Risk Ratio [RR] 24.82, 95%CI 21.12 to 29.16; RR 2.207, 95%CI 1.427 to 3.412; and RR 7.900, 95%CI 5.386 to 11.578, respectively) and in the general population (RR 23.614, 95%CI 20.584 to 27.129; RR 3.103, 95%CI 2.056 to 4.685; and RR 49.500, 95%CI 12.256 to 199.921, respectively). In conclusion, our study stresses the possibly high underestimation of OROV prevalence in the general population of South America, the potential global threat represented by this arbovirus infection, and the potential preventive role of a comprehensive “One Health approach”.

## 1. Introduction

Oropouche Virus (OROV) is an enveloped RNA virus, sized from 80 to 120 nm in diameter, that belongs to the serogroup Simbu of the viral genus *Orthobunyavirus* [1,2,3], the largest of the five genera of the Peribunyaviridae family in the order Bunyavirales [4], and causes Oropouche Fever (OF), a febrile illness whose clinical features overlap with infections by other arboviruses such as Dengue (DENV), West Nile Virus (WNV), Yellow Fever Virus (YFV), Zika Virus (ZIKV), and Chikungunya Virus (CHKV) [1,5,6]. The OROV genome comprises three single-stranded, negative-sense RNA segments, designated as Large (L), Medium (M), and Small (S) [1,2,4,7]. The L segment mainly encodes for RdRp, a polymerase involved in the replication of the viral genome within the cytoplasm of infected cells [1,7]. Unsurprisingly, the L segment is associated with the highest homology among sampled strains, close to 100%. The main antigens (glycoproteins Gn and Gc), alongside the non-structural protein m (NSm), are encoded by the M segment. Finally, the S segment encodes for both the nucleocapsid protein (N) and non-structural protein s (NSs) from overlapping open reading frames [4]. The S segment is usually associated with the highest heterogeneity in terms of gene sequencing, and not coincidentally, the protein N is the cornerstone for the classification of OROV into four genotypes (I, II, III, and IV) [2,4,8].

According to our current understanding [8], OROV was initially identified in 1955 in a febrile worker from the Melajo Forest of Trinidad and Tobago during an outbreak of febrile illness [9,10,11,12], and neutralizing antibodies were also detected both in their colleagues and in a local monkey population. Since then, epidemics have occurred in several countries in South and Central America [1,6,7,8], starting from an outbreak that occurred in Belem, in the Brazilian state of Para, in 1961 [10,11,12,13,14,15], eventually accounting for around half million cases in the following decades. At the same time, OROV silently spread from the Amazon region to other areas of Brazil, extending to countries across Central and South America, including Argentina, Colombia, Bolivia, Ecuador, French Guyana, Panama, Peru, and Venezuela [3,16,17,18,19,20,21,22,23,24], and eventually returning to the Caribbean, with a recent outbreak in the island of Cuba [23,25,26].

In OROV, interhuman spread has not been reported; the virus is transmitted to humans through the bite of infected insects, including *Culicoides paraensis* midges (predominant in urban settings) and *Culex quinquefasciatus* mosquitoes, but other arthropod species have also been documented as harboring the pathogen (e.g., *Aedes serratus*, *Coquillettidia venezuelensis*, and other *Culicoides* species) [1,5,6,7,27,28]. While the importance of *Culicoides paraensis* has been documented since the early study by Pinheiro et al. in 1982 [29], the primary vertebrate host of OROV, if any, has not been reported. In fact, known vectors feed on and take their blood meals from an extensive range of vertebrates, ranging from the three-toed sloth (*Bradypus tridactylus*) to non-human primates, such as the capuchin and howler monkeys [1,2,3,7,17]. While *C paraensis* is commonly found in water bodies (i.e., ponds, lakes, and rivers) from humid tropical regions, being the predominant vector in the sylvatic environment [7,27], both *C paraensis* and *Culex quinquefasciatus* thrive in areas characterized by very poor living standards, equally contributing to urban outbreaks [5]. Moreover, Culicoides are frequently associated with banana or cacao farming, where the Culicoides breed in discarded husks and banana stalks. Considering environmental (i.e., deforestation and disorganized urbanization) and climate changes that collectively contribute to vector proliferation and the landscape modeled by human migrations and global travels (which in turn contribute to virus dissemination) [2,30], the likelihood of new OROV outbreaks and its eventual spread outside South America would be far from surprising [1,2,3,7,25,30,31,32,33,34,35]. In fact, 2024 has been associated with a sustained increase in reported cases compared with 2023 in the whole of South America, as by 9 May 2024, a total of 4583 cases had been reported in Brazil compared with the 835 positive cases in the whole of 2023 [36,37], and these figures did further increase in the following months, with a total of 8078 confirmed cases of OROV infection by 30 July 2024 (7284 cases from Brazil, 356 cases from Bolivia, 290 from Peru, and 74 from Colombia and Cuba) [38].

The features of OF have been usually described as a clinical syndrome characterized by fever, headache, muscle pain, and joint pain that usually manifests three to eight days after being bitten by an OROV-infected vector and lasts three (usual range, two to seven) days, then recurring one to two weeks after the initial recovery in around 60% of affected individuals [2,5,31,35,39]. Complications such as hemorrhagic phenomena (i.e., epistaxis, gingival bleeding, and petechiae) and the involvement of the central nervous system (i.e., meningitis, encephalitis, dizziness, and anorexia) have been reported, but their actual occurrence is hardly defined due to the uncertainties about the epidemiology of OROV [8,13,40,41,42,43,44]. In fact, until 2024, OF was consistently considered a self-limited and mostly benign condition, as no deaths and long-term sequelae were associated with OROV infection. Conversely, during the outbreak of 2024, the Brazilian Ministry of Health released a report on four cases of microcephaly in newborns from infected mothers, and by late July, two deaths were reported [38,45,46]. However, the overlap of clinical features with those associated with other arboviruses poses a significant challenge for an accurate diagnosis, as laboratory confirmation is mandatory but not consistently performed [6,47], particularly during the ongoing Dengue epidemic in South America [6].

In summary, the emergence (or, more appropriately, the re-emergence) of OROV in South America, with the potential of the further spread of this pathogen even in other continents [26,48], highlights the importance of gathering and critically appraise available evidence on the current epidemiology of OROV infection, which will be reported in the present systematic review with meta-analysis.

## 2. Materials and Methods

A systematic review and meta-analysis were designed in accordance with the PRISMA statement (Prepared Items for Systematic Reviews and Meta-Analysis) [49,50]. The study outline was preventively recorded in the PROSPERO (Prospective Register of Systematic Reviews) database with ID number CRD42024576181 (Appendix A).

### 2.1. Research Concept

The research concept was defined according to the “PICO” strategy (Patient/Population/Problem, Investigated results, Control/Comparator, and Outcome) as outlined below. 

Population of interest: general population.

Investigated result: prevalence of biomarkers for current and previous exposure to or infection by Oropouche Virus (i.e., detection of viral antigens, IgG/IgM antibodies, viral copies, and/or viral RNA).

Control: occurrence of arboviral infections (i.e., Dengue, Chikungunya, and Zika in the same population groups).

Outcome: prevalence of Oropouche infection in the general population.

### 2.2. Study Selection and Inclusion and Exclusion Criteria

Starting on 3 August 2024, three scientific databases (i.e., PubMed, EMBASE, and Scopus) and the preprint repository MedRxiv were searched for entries on OROV and OF without any backward chronological restriction. The research strategy was adapted to the specificities of the inquired databases, and it is summarized in Appendix B Table A1. Moreover, a “snowball” approach was applied, with references to the retrieved studies being accurately searched for further suitable entries. For the aims of this review, observational studies were only considered suitable if written in English, Italian, German, French, Spanish, and Portuguese. The retrieved articles were initially assessed through title screening for their relevance to the subject [49,50]. When an article was positively title-screened, the content of the abstract was then screened, and if considered consistent with the aims of the present review, the full text was retrieved and independently assessed by two investigators (M.B. and S.C.).

To be considered consistent with this review and therefore included in the present systematic review, the following inclusion criteria had to be fulfilled:(1)Availability of the full text.(2)The selection criteria for tested participants were preventively reported; for the aims of the present review, both studies on the whole of the resident population from a certain area and studies focusing on patients with potential signs and symptoms were considered.(3)OROV infection was assessed by means of (a) detection of OROV and/or its RNA in blood, saliva, and/or cerebrospinal fluid (CSF) or central nervous system (CNS) tissue; (b) detection of OROV antigens by immunohistochemical analysis with virus-specific monoclonal antibodies; (c) detection of OROV-reactive IgG and/or IgM in a serum or CSF sample with and without confirmation assay detecting antibodies against individual OROV antigens.

Moreover, the following exclusion criteria were applied:(1)Derivative studies (i.e., systematic reviews and meta-analyses), letters, editorial comments, and case reports;(2)Studies on animals (including non-human primates);(3)The full text was not available either through online repositories or through inter-library loans, or its main text was written in a language different from English, Italian, German, French, Spanish, or Portuguese;(4)Lack of details about geographical setting and corresponding timeframe.

### 2.3. Data Extraction

The data extracted included the following (where available):(a)Settings of the cases (year, month, or season and geographic region);(b)Recruitment strategy, summarized as a dichotomous variable: asymptomatic vs. symptomatic subjects;(c)Laboratory testing strategy, summarized as virological studies (i.e., all laboratory testing leading to the isolation of OROV or its RNA through viral cultures and/or real-time quantitative polymerase chain reaction studies), antigenic studies (i.e., all studies leading to the identification of OROV antigens), and serological studies (i.e., all studies leading to the identification OROV-reactive antibodies).(d)Where available, data on the testing of other arboviruses (i.e., DENV, CHKV, and ZIKV) were similarly collected.

If case series were cross-posted by different studies, reports were accurately analyzed to fill the knowledge gaps, provide a more extensive description of clinical cases, and eliminate duplicates.

### 2.4. Qualitative Assessment

The risk of bias (ROB) has been defined as the likelihood that any feature from the design or conduct of the study may lead to misleading results [51,52,53]. As a consequence, ROB assessment helps to establish transparency of evidence from the findings. For the aims of the present systematic review, ROB was assessed by means of the ROB tool from the National Toxicology Program (NTP)’s Office of Health Assessment and Translation (OHAT) (now Health Assessment and Translation (HAT) group) [53,54]. The OHAT ROB approach for observational studies starts with 7 questions summarized in 6 domains, with each domain being rated on a 4-point scale (“definitely low”, “probably low”, “probably high”, and “definitely high”): participant selection (D1), confounding factors (D2), attrition/exclusion (D3), detection (D4), selective reporting (D5), and other sources of bias (D6). In this study, we did prioritize OHAT ROB over other similar instruments as it does not require that studies affected by a certain degree of ROB be removed from the pooled analyses [54], as even studies affected by a higher ROB may provide important information and can be used in sensitivity analysis.

Two investigators independently rated all articles according to the current indications. Their disagreements were primarily resolved by consensus between the two reviewers; when consensus was not possible, input from a third investigator (M.R.) was requested and obtained.

### 2.5. Data Analysis

#### 2.5.1. Descriptive Analysis

All included studies were initially summarized by descriptive analysis. Therefore, crude prevalence figures for the OROV detection rate per 100 people were calculated by specimen (e.g., blood/serum, salivary specimens, and CSF), testing strategy (e.g., studies targeting viral identification, studies targeting viral antigens, and studies targeting viral antibodies), and sampling group (e.g., symptomatic vs. asymptomatic people). 

#### 2.5.2. Meta-Analysis

Pooled estimates of OROV prevalence were calculated through a random effect model (REM) meta-analysis and reported as point estimates with their 95% confidence intervals (95%CIs). For the aims of our study, we prioritized a REM approach over a fixed effects model, as in the REM, each study estimates a different underlying actual effect, and these effects have a distribution across the levels of the variable. In other words, as random effects account for variability and differences between different entities or subjects within a larger group, the REM is usually considered more effective in dealing with the genuine differences underlying the results of the studies or heterogeneity [55,56]. 

#### 2.5.3. Heterogeneity

Heterogeneity has been defined as the inconsistency of effect among the included studies and can be considered by means of the I^2^ statistic as the percentage of total variation across the included studies likely occurring because of actual differences rather than chance [51]. For the aims of the following studies, heterogeneity was considered low for I^2^ values ranging from 0 to 25%, moderate for I^2^ values ranging from 26% to 50%, and substantial for I^2^ values ≥ 50%. As suggested by Hippel et al. [51], for providing a more appropriate reporting of actual heterogeneity, the 95%CIs of the I^2^ estimates were calculated and reported. 

#### 2.5.4. Sensitivity Analysis

Sensitivity analysis is the study of how the uncertainty in the output of a mathematical model or system can be apportioned to different sources of uncertainty in its inputs. In the present systematic review and meta-analysis, the effect of each study on the pooled estimates was estimated by calculating each estimate by excluding one study at a time.

#### 2.5.5. Publication Bias

Publication bias (i.e., the likelihood of a study being published based on the direction of the findings) was initially assessed through the calculation of funnel plots. Funnel plots are simple scatter plots of the effect estimates from individual studies plotted on the horizontal axis against the standard error of the estimated effect on the vertical axis. As a preliminary step, the asymmetry of funnel plots was visually assessed; in the presence of bias, an asymmetrical appearance to the funnel will emerge because small studies without statistically significant effects are potentially unpublished, and greater asymmetry will suggest more significant bias. Visual interpretation of the plots was assisted by implementing contours of statistical significance. The asymmetry of funnel plot outcomes with three or more included studies was then assessed by means of Egger’s test [49,57]. Small-study bias was eventually assessed by means of radial plots (i.e., a graphical display for comparing estimates that have differing precisions). A *p*-value < 0.05 was considered statistically significant for both publication and small-study bias.

#### 2.5.6. Software

The screening of the retrieved articles was performed on Mendeley Reference Manager (version 2.121.0; Mendeley Ltd.; New York, NY, USA). All calculations were performed by means of R (version 4.4.1) [58] and Rstudio (version 2024.04.2 Build 764; Posit Software, PBC; Boston, MA, USA) software by means of the packages meta (version 7.0), fmsb (version 0.7.5), “epiR” (version 2.0.63), and “robvis” (version 0.3.0). Plots were calculated by means of the R packages “ggplot2” (version 3.4.3), “ggpubr” (version 0.6.0), and “PRISMA2020” (version 1.1.1) and GraphPad Prism, version 10.0 (GraphPad Software LLC, Boston, MA, USA). A Prisma2020 flow diagram was designed by means of the PRISMA2020 package [59].

## 3. Results

### 3.1. Search and Selection Process

A total of 745 entries were retrieved from the four inquired databases (i.e., 239 from Pubmed, 32.08%; 215 from EMBASE, 28.86%; 280 from SCOPUS, 37.58%; and 11 from medRxiv, 1.48%); of these, 434 were duplicated across the databases and were removed from the analyses (58.26%). Moreover, 210 of the 311 records screened by title and abstract were removed, as they were not consistent with the research aims (28.19%); regarding the 101 articles sought for retrieval, all of them were eventually retrieved and individually assessed in their full text. A total of 57 articles (7.65%) were removed, as they were not consistent with the inclusion criteria, leaving 44 articles (5.91%). However, the citation analysis led to the identification of 3 further studies that were included in the pool of sample entries, for a total of 47 observational studies (Figure 1).

### 3.2. Summary of Included Studies

The 47 retrieved studies are summarized in Table 1 and Figure 2 [10,11,12,13,15,16,19,22,23,24,32,34,39,40,41,47,60,61,62,63,64,65,66,67,68,69,70,71,72,73,74,75,76,77,78,79,80,81,82,83,84,85,86,87,88,89,90]. Briefly, the reported studies were published from 1976 [10] to 2024 [41,86,87,88,89,90], encompassing 91 series (i.e., groups of sampled individuals taken from various temporal and geographic settings), for a total of 62,827 blood samples [10,11,12,13,15,16,19,22,23,24,32,34,39,40,41,47,61,62,63,64,65,66,67,68,69,70,73,74,75,76,77,78,79,80,81,82,83,84,85,86,87,88,89,90] and 309 samples of cerebrospinal fluid (CSF) [60,71,72]. All series but one (i.e., Haiti, *n* = 1250 samples; 1.99% of the total blood samples) were from South America: Bolivia (4 series, 2089 blood samples; 3.33%) [24], Brazil (51 series, 22,744 blood samples; 36.20% of the total; the whole of CSF specimens) [10,11,12,13,19,39,40,41,60,61,62,64,65,66,67,68,71,72,73,77,79,80,81,82,85,86,87,88,89,91], Colombia (10 series, 3747 blood samples; 5.96%) [19,32,84,90], Ecuador (3 series, 883 blood samples; 1.41%) [22,24], French Guyana (1 series, 95 blood samples; 0.15%) [16], Paraguay (2 series, 340 blood samples; 0.54%) [24,47], and Peru (28 series, 31,988 blood samples; 50.91%) [13,14,15,34,63,69,70,75,76,78,85].

The specimens were collected from subjects either reporting any sign or symptom of arboviral infections (all CSF specimens and 57,108 blood specimens, 90.90%) or the general population (5719, 9.10%). Focusing on studies including individuals with any sign or symptom, in most cases, the index patients did report acute febrile illness (fever > 38°) lasting 5 to 7 days and were therefore suspected of underlying arboviral infection, with a further study including individuals with cutaneous rash (340 samples, 0.54% of the total) [85]. Interestingly, in all series based on CSF samples, the sampled patients had a previous diagnosis of CNS infection. Regarding the testing strategy, 32,660 (52.00% of total samples) of the collected blood specimens targeted viral particles by means of either viral isolation studies (924 samples, 1.47%) [10,61,75,82] or viral RNA through RT-qPCR (31,726 samples, 50.50% of the total) [16,22,23,24,32,34,41,47,66,73,74,75,76,77,78,79,80,83,85,86,87,88,90], and in 20,975 samples (33.39%), the methods used were the confirmatory test of IFAT (20,880 samples, 33.23%) [24] or the microneutralization test (95 samples, 0.15%) [16]. In all studies on CSF, viral particles were targeted either by means of viral isolation [60] or RT-qPCR [71,72]. The serological studies included a total of 24,448 specimens (38.91% of total samples); of these, 6607 in a single study by the Loreto department in Peru were tested by means of IFAT (10.51%) [15], 3194 (5.08% of total blood samples) were analyzed by means of the hemagglutination test [10,11,12,67] between 1976 and 2009, and 1610 (2.56%) were analyzed by means of the plaque reduction neutralization test [66,70,84,86]. All remaining samples were tested by means of ELISA, either as a single testing option (1404 samples, 2.23%) or as the confirmatory test for a hemagglutination test (11,329 samples, 18.03%). At the same time, all tests based on ELISA reported on the occurrence of IgM class antibodies, while all other studies were unable to discriminate between the occurrence of IgM and/or IgG antibodies and were considered in terms of IgM and/or IgG status (IgM/IgG). By the time of our systematic review and meta-analysis (summer 2024), no viral antigen-based studies had been identified.

Focusing on data from the general population, specimens were collected from 21 series included in eight studies [13,14,19,62,65,81,84,89]. Recruitment strategies were quite heterogeneous. While Rosa et al. [62] and Watts et al. [14] performed random sampling among the targeted population, Grisales-Nieto et al. [19] only included individuals who were previously sampled as negative for other arboviral studies. In turn, Baisley et al. [13], Silva et al. [89], and Gil-Mora et al. [84] tentatively sampled all individuals from the general population of the inquired area, excluding younger age groups, except for the study by Gil-Mora et al. [84], which included all subjects aged less than 18 years at the time of the survey. Eventually, Tavares-Neto et al. [65] randomly sampled individuals receiving the Hepatitis B Virus vaccine during a vaccination campaign, while Salgado et al. [81] reported on a sample from the Brazilian armed forces. Moreover, the aforementioned study by Watts et al. [14] performed serial sampling among participants, with a follow-up one year after the preliminary study among individuals who were initially characterized as seronegative cases.

A total of 5248 blood specimens (8.35% of total samples) targeted OROV serology by means of the hemagglutination test (298, 0.47%), the immunochromatographic test (410, 0.65%), the plaque reduction neutralization test (455, 0.72%), and mainly ELISA, either as a single testing strategy (2168, 3.45%) or as the confirmatory test for a previous hemagglutination test (1917, 3.05%). Interestingly enough, only two studies, for a total of 599 samples, specifically targeted IgM class antibodies (0.95% of total blood samples), compared with 3896 samples specifically targeting IgG class antibodies (6.20%). In contrast, both the hemagglutination and plaque reduction neutralization tests (753 samples, 1.20%) were by their design unable to characterize the detected antibodies as IgG or IgM and were therefore considered in terms of IgM and/or IgG status (IgM/IgG). Finally, only two studies, for a total of 471 samples (0.75% of total blood samples), aimed at virus identification by means of either virus isolation [62] or viral RNA [19]. No viral antigen-based studies were performed in the general population.

In most studies, both in symptomatic patients and the general population, OROV was the only arbovirus tentatively detected. However, focusing on the former, paired testing for DENV was performed in 17 studies (including 34 series and 46,537 specimens) [15,22,23,24,32,41,47,64,67,72,73,76,79,80,82,84,85], with 9 studies (9 series and 3130 specimens) performing paired testing for CHKV [23,41,76,79,82,84,85,86] and 6 studies (6 series and 2758 specimens) for ZIKV [23,41,76,79,82,85]. Furthermore, four studies [65,81,84,89], including data from the general population, reported on paired tests for DENV (1557 samples), ZIKV (708 samples), and CHKV (1162 samples). 

### 3.3. Risk of Bias Analysis

The overall quality of the included studies is summarized in Figure 3, while a detailed report for each study is provided in Appendix B Table A2. Briefly, mixed quality was documented, and the main issue shared by most of the included studies can be related to the selection bias (D1), particularly when dealing with prevalence studies based on individuals reporting any sign or symptom allegedly associated with arbovirus infection. This potential shortcoming was particularly high for the study by Sanchez-Lerma et al. [90], as the study only included 100 consecutive cases; it was, therefore, impossible to ascertain the sample’s representativity. Similarly, due to the inclusion of early studies from the 1970s, the laboratory testing strategy was possibly affected by substantial uncertainties (D2), particularly when dealing with seroprevalence studies, as hemagglutination tests and plaque reduction neutralization tests were unable to dichotomize between IgM and IgG class antibodies, i.e., recent vs. previous OROV infection. Finally, the studies by Salgado et al. [81] and Elbadry et al. [23] were not only heterogeneous in terms of geographic settings, with the former study being in the Caribbean rather than South America and the latter study adopting nationwide sampling (being thus hardly comparable to other regional-based settings), but also included two very selective population groups (i.e., students and military professionals, respectively), whose exposure and clinical baseline features were in turn very distinctive from other groups.

### 3.4. Summary of Main Results

#### 3.4.1. Crude Prevalence Estimates

##### OROV Identification

As shown in Table 2, after the removal of duplicate isolates from the retrieved series, a pooled sample of 32,439 specimens tested for OROV was identified, and a crude detection rate of 1.64% was calculated. The majority of samples were collected between 1991 and 2010 (21,178, 65.29%), with the lowest detection rate (0.09%), compared with 26.74% of the 344 samples taken before 1990 and the 3.86% of the 10,917 samples collected after 2010. Assuming the studies performed before 1990 (all of them based on direct virus identification) as the reference group, Risk Ratios (RRs) of 0.003 (95%CI 0.002 to 0.005) for 1991–2010 and 0.144 (95%CI 0.118 to 0.175) after 2010 were identified.

The large majority of the sampled cases were from Peru (*n* = 20,989; 64.70%), followed by Brazil (4030, 12.42%), Colombia (3067, 9.45%), Bolivia (2089, 6.44%), Haiti (1250, 3.85%), Ecuador (579, 1.78%), Paraguay (340, 1.05%), and French Guyana (95, 0.29%), where the latter country was also characterized by the highest detection rate (24.21%), with a corresponding RR of 6.023 (95%CI 4.092 to 8.864) compared with Brazil. On the contrary, the detection of OROV-positive specimens was substantially lower for Peru (1.15%; RR 0.286, 95%CI 0.235 to 0.348) and Haiti (0.08%; RR 0.020, 95%CI 0.003 to 0.142), and it was comparable to that of Brazil for Colombia (3.42%; RR 0.852, 95%CI 0.669 to 1.084), while no positive samples were reported from Paraguay, Bolivia, and Ecuador.

The overwhelming majority of tests were performed on blood specimens (32,152; 99.12%), with a detection rate of 1.63%, compared with 2.44% among the 287 CSF specimens (RR 1.494, 95%CI 0.715 to 3.120). When taking into account laboratory testing techniques, RT-qPCR was associated with a detection rate of 3.36%. Assuming RT-qPCR as the reference group, the highest detection rate was associated with a testing strategy where RT-qPCR was the confirmatory test for a microneutralization assay (24.21%; RR 7.214, 95%CI 4.981; 10.446), followed by virus isolation (14.59%; RR 4.347, 95%CI 3.612; 5.231), while the lowest detection rate was associated with a testing strategy where RT-qPCR was performed on positive IFAT specimens (0.08%, RR 0.026, 95%CI 0.016 to 0.041). 

##### Serology of OROV Infection

Data on the serology of cases with any sign and/or symptom of arbovirus infection (*n* = 24,270) are reported in Table 3 after the removal of duplicated series. A cumulative detection rate of 6.66% was calculated. More precisely, the large majority of samples were collected between 1991 and 2010 (14,192, 58.48%), followed by the timeframe after 2010 (8462, 34.58%) and the smallest group, including samples collected before 1990 (1816, 7.42%). Assuming the latter group as the reference one, both the former timeframes were associated with a substantially decreased risk for the detection OROV targeting antibodies (RR 0.108, 95%CI 0.097 to 0.120, and RR 0.128, 95%CI 0.114 to 0.143, for studies performed between 1991 and 2010 and after 2010, respectively).

The majority of samples were retrieved from Brazil (15,973, 64.54%), followed by Peru (8323, 34.01%), Bolivia (2089, 8.54%), Ecuador (304, 1.24%), and Colombia (50, 0.20%). Assuming Brazil as the reference group (7.45%), substantially lower detection rates were associated with Peru (5.43%; RR 0.729, 95%CI 0.656 to 0.809) and Ecuador (0.33%, RR 0.044, 95%CI 0.006 to 0.313), while no positive cases were found in Bolivia and Colombia.

The overwhelming majority of samples included blood samples (24,448, 99.91%) with only 22 specimens of CSF. The corresponding detection rates were 6.65% and 13.64%, but the difference was not statistically significant (RR 2.049, 95%CI 0.715 to 5.871).

Regarding the targeted antibodies, most samples did not discriminate between IgM and IgG classes (53.25%), while the remaining studies calculated the occurrence of IgM class antibodies. The corresponding detection rates were 6.89% and 6.40%, respectively, with no substantial differences (RR 1.077, 95%CI 0.980 to 1.184). Regarding the diagnostic procedure, most of the cases were analyzed by means of ELISA (13,037; 53.28%), followed by IFAT (6607, 27.00%), hemagglutination test (3216, 13.14%), and plaque reduction neutralization test (1610, 6.58%), with corresponding detection rates of 5.79%, 1.03%, 19.59%, and 10.99%. Taking into account the detection rate documented by ELISA as the reference group, the occurrence was substantially higher for studies based on the hemagglutination test (RR 3.383, 95%CI 3.065 to 3.733) and plaque reduction neutralization (RR 1.898, 95%CI 1.625 to 2.217), with a reduced rate for IFAT (RR 0.178, 95%CI 0.139 to 0.227).

Seroprevalence data from the general population are reported in Table 4. More precisely, a total of 5247 samples were analyzed, with a crude detection rate of 21.21%. All samples were collected after 1990, with 4085 cases being from the timeframe 1991–2010 (77.85%) and 1162 after 2010 (22.15%). The corresponding detection rates were 26.95% and 1.03%, respectively, with a reduced risk for specimens collected after 2010 (RR 0.038, 95%CI 0.022 to 0.067).

The studies were performed only in Brazil (2625 samples, 50.03% of cases), Peru (2168 samples, 41.32%), and Colombia (455, 8.67%), with corresponding detection rates of 24.61%, 21.08%, and 2.20%. Assuming Brazil as the reference group, the risk for positive serostatus was significantly lower in both Peru (RR 0.857, 95%CI 0.771 to 0.952) and Colombia (RR 0.089, 95%CI 0.048 to 0.165).

The majority of samples were analyzed by means of ELISA (4085, 77.85%), followed by plaque reduction neutralization test (455, 8.67%), immunochromatographic test (410, 7.81%), and hemagglutination test (298, 5.68%). The corresponding detection rates were 26.95% (reference group), 2.20% (RR 0.082, 95%CI 0.044 to 0.151), and 0.67% (RR 0.025, 95%CI 0.006 to 0.099) with no cases detected with immunochromatographic testing (RR 0.005, 95%CI 0.001 to 0.072). 

Regarding the targeted classes of antibodies, most samples specifically identified OROV targeting IgG (3896, 74.25%), with 599 samples reporting the occurrence of OROV targeting IgM (11.42%) and 753 samples not discriminating between IgG and IgM antibodies (14.35%). The corresponding prevalence rates were 28.03%, 1.50%, and 1.59%. Assuming the seroprevalence of IgM class antibodies as the reference group, a substantially higher rate for IgG antibodies was documented (RR 18.655, 95%CI 9.735 to 35.746), with comparable estimates for the studies unable to discriminate between IgG and IgM (RR 1.061, 95%CI 0.450 to 2.500).

##### Occurrence of Other Arboviral Infections

Table 5 reports on the detection rates for other pathogens in paired samples from the included studies. The detection rates for all sampled pathogens were substantially higher than that for OROV. More precisely, RRs of 24.816 (95%CI 21.119 to 29.159) and 23.631 (95%CI 20.584 to 27.129) were identified for DENV Virus identification and serology, respectively, compared with an RR of 7.900 (95%CI 5.386 to 11.578) and an RR of 49.500 (95%CI 12.256 to 199.921) for ZIKV and an RR of 2.207 (95%CI 1.427 to 3.412) and an RR of 3.103 (95%CI 2.056 to 4.685) for CHKV.

#### 3.4.2. Meta-Analysis of Prevalence Estimates

##### Detection Rates among Individuals with Signs/Symptoms of Arbovirus Infection

A REM meta-analysis was performed, and the main results for blood specimens are reported in Table 6, while the corresponding forest plots are reported in Appendix B Figure A1, Figure A2 and Figure A3. Briefly, the detection rate for the whole of viral isolates was 0.45% (95%CI 0.16 to 1.23), with substantial heterogeneity (I^2^ 94.9%, 95%CI 85.7 to 93.9). In subgroup analysis, the detection rate for direct virus isolation was 13.91% (95%CI 5.66 to 30.34, I^2^ = 95.3), compared with 0.80% (95%CI 0.31 to 2.06) for RT-qPCR as the sole detection test and 0.03% (95%CI 0.01 to 0.16, I^2^ 93.5%) for RT-qPCR as the confirmatory test for IFAT. The estimate for RT-qPCR as the confirmatory test for a microneutralization test was 24.21% (95% 16.65 to 33.81), but no I^2^ statistics were calculated, as only one estimate was included.

Regarding serological estimates, the overall detection rate for IgM was 12.45% (95%CI 3.28 to 37.39), with substantial heterogeneity (I2 98.5%, 95%CI 98.1 to 98.8%). Studies based on ELISA as the sole testing option had an estimated detection rate of 34.83% (95%CI 20.42 to 52.69; I^2^ 97.4%), compared with 6.61% (95%CI 0.95 to 34.27; I^2^ 98.8%) for studies having ELISA as the confirmatory test for a hemagglutination test. Finally, the detection rates for tests unable to discriminate between IgM and IgG classes was 12.21% (95%CI 4.96 to 27.09), with substantial heterogeneity (I^2^ 99.0%, 95%CI 98.8 to 99.2), encompassing 24.66% (95%CI 10.97 to 46.51, I^2^ 98.7%) from studies based on hemagglutination tests, 4.31% (95%CI 1.12 to 15.14; I^2^ 90.6%) from studies based on plaque reduction neutralization test, and 1.03% (95%CI to 1.30) from the single study based on IFAT.

As shown in Figure 4, the pooled estimate for OROV detection in CNS specimens was 2.40% (95%CI 1.68, 5.03), with low heterogeneity (I^2^ 0%, 95%CI 0.0 to 89.6%).

##### Detection Rates in General Population

As only two series included data from viral isolates [19,62], and similarly, two series reported data on IgM [65,89], a REM meta-analysis was performed only for detection rates of IgG and for studies unable to discriminate between IgG and IgM. More precisely, the pooled detection rate for the whole of IgM class antibodies was 24.45% (95%CI 7.83 to 55.21), with substantial heterogeneity (96.6%, 95%CI 96.5 to 98.3), encompassing estimates of 59.82% (22.71 to 88.29, I^2^ 99.5) for studies based on ELISA only and 29.31% (95%CI 21.91 to 37.99, I^2^ 92.6%) for studies where ELISA was a confirmatory test for a hemagglutination test, and the single estimate from the series based on immunochromatography, with no positive specimens (Table 7).

Considering the detection rates from studies unable to discriminate between IgM and IgG, a pooled estimate of 1.42% (95%CI 0.61 to 3.28), with substantial heterogeneity (I^2^ 41.9%, 95%CI 0.0 to 80.5), was calculated, encompassing a prevalence estimate of 2.16% (95%CI 0.99 to 4.68, I^2^ 6.2%) and the single estimate for the hemagglutination test.

As only two studies documented viral isolates, by means of either RT-qPCR (estimated detection rate for OROV: 0.57%) [19] or virus isolation (estimated detection rate: 3.38%) [62], a meta-analysis of the corresponding pooled estimates was not performed.

### 3.5. Sensitivity Analysis

A sensitivity analysis (i.e., the study of how different values of an independent variable affect a dependent variable under a given set of assumptions) was performed by removing from the REM meta-analysis one single individual series at a time; the resulting pooled estimates are reported as Appendix B Figure A6, Figure A7, Figure A8, Figure A9, Figure A10 and Figure A11. 

#### 3.5.1. Studies on Individuals Affected by Any Sign or Symptom of Arboviral Infection, Seroprevalence Studies

As highly expected due to the heterogeneity of the included studies, estimates on IgM prevalence were highly affected by the removal of individual studies (Appendix B Figure A6), and more precisely the series by Figuereido et al. [64], Manock et al. [22], and Cruz et al. [67]. Notably, the estimated detection rates for OROV-targeting IgM class antibodies increased from 12.45% (95%CI 3.28 to 37.39) to 20.17% (95%CI 8.11 to 41.98), 16.58% (95%CI 4.73 to 44.33), and 14.94% (3.76 to 44.11), respectively. On the contrary, while pooled heterogeneity remained substantially unaffected, the estimate for between-study variance (tau^2^) was particularly affected by the removal of the study by Figuereido et al. [64], dropping from 6.310 to 3.091. Similarly, estimates from studies on the occurrence of OROV-targeting antibodies not discriminating between IgM and IgG classes (Appendix B Figure A7) were affected by the removal of the series by Gil-Mora et al. [84], Watts et al. [15], and Bernardes Terzian et al. [66], with prevalence estimates increasing from 12.21% (95%CI 4.95 to 27.09) to 14.67% (95% 6.31 to 30.48), 14.86% (95%CI 6.31 to 31.14), and 14.19% (5.78 to 30.84). Notably, the removal of two series from a single study by Pinheiro et al. [10] impacted the pooled estimates oppositely, as the removal of the cases from the village of Mojuì from early 1976 decreased the estimated detection rate to 9.82% (95%CI 4.25 to 21.06), while the removal of other villages sampled in the following months (February to May 1975) increased the estimated detection rate to 13.69% (95%CI 5.49 to 30.22). On the contrary, the removal of single estimates did not impact on the heterogeneity of the pooled estimates.

#### 3.5.2. Studies on Individuals Affected by Any Sign or Symptom of Arboviral Infection, OROV Detection

As shown in Appendix B Figure A8, the removal of individual series did not impact the pooled estimates.

#### 3.5.3. Studies on General Population— Seroprevalence Estimates

As shown in Appendix B Figure A9, the removal of individual series also impacted the pooled estimates. More precisely, the removal of the samples from the village of El Tambo included in the study by Gil-Mora et al. [84] reduced the detection rate from 1.42% (95%CI 0.61 to 3.28) to 0.82% (95% 0.31 to 2.15), while omitting the report by Salgado et al. [81] increased the estimated detection rate to 2.16% (95%CI 0.98 to 4.68); interestingly enough, both series negatively impacted the heterogeneity of the estimates, as their removal reduced the pooled I^2^ to 0% and 6%, respectively. On the contrary, when dealing with the detection rate of IgG class antibodies (Appendix B Figure A10), only two series scored a substantial impact on the pooled estimates, as the reported detection rate of 24.45% (95%CI 7.83 to 55.21) shifted to 18.53% (95%CI 6.22 to 43.82) and 37.57% (95%CI 23.19 to 54.53) following the removal of Rosa et al. [62] and Silva et al. [89], respectively. Notably, only the removal of the former study affected the pooled I^2^ estimates, which dropped from 98% to 90%.

#### 3.5.4. Studies on Detection of OROV in CNS

The sensitivity analysis for studies on CSF is reported in Appendix B Figure A11. Taking into account the reduced number of included studies (only three), the removal of the report by Bastos et al. [72] increased the estimated detection rate from 2.44% (95%CI 1.17 to 5.03) to 3.28% (95%CI 1.24 to 8.41).

### 3.6. Publication Bias

Publication bias was ascertained by calculating funnel plots for all pooled estimates with their subsequent visual inspection. In a funnel plot, the sample size is plotted against the effect size (i.e., detection rate), and as the size of the sample increases, the individual estimates of the effect likely converge around the true underlying estimate [62,65,72]. The funnel plots for prevalence estimates are reported in Appendix B Figure A12.

Due to the uneven sample size of all estimates, the funnel plots were highly asymmetrical, and this finding stresses the likelihood of publication bias. However, as shown in Table 8, Egger’s test (i.e., a linear regression of the effect estimates on their standard errors weighted by their inverse variance), on the contrary, only reported substantial asymmetry for studies on OROV detection (t -4.37, bias -3.745, *p* < 0.001). Therefore, other potential sources of bias could explain the reported asymmetry, including the heterogeneous choices in the outcome measure and, most notably, the differences in the underlying risk, as for studies performed in settings characterized by heterogeneous incidence of the pathogen.

The analysis of small-study bias by means of radial plots (scatter plots of standardized estimates) is shown in Appendix B Figure A13. In fact, all plots were characterized by an uneven distribution of the point estimates across the regression lines, suggesting that all estimates may have been somehow affected by smaller samples.

## 4. Discussion

### 4.1. Synthesis of Main Findings

Our systematic review with meta-analysis on the epidemiology of OROV and OF gathered a total of 47 observational studies spanning from 1975 to 2024. Most of the available evidence was collected from studies performed in South America, with a further report from the Caribbean. The collected studies were either performed on symptomatic individuals during an outbreak or as prevalence studies from the general population of areas with either documented or suspected occurrence of OROV infection and also in countries (i.e., Paraguay) where no documented OROV epidemics have occurred at least up to the present [93]. In turn, both strategies were quite heterogeneous in terms of the inquired biomarkers. A total of 33,156 individuals with signs and/or symptoms of arbovirus infection were sampled for the occurrence of OROV, either by virus isolation or by means of RT-qPCR, which was either performed on the whole of the sample or among patients where OROV infection was preventively suspected by serological tests. Not coincidentally, the estimated prevalence ranged from 13.91% (95%CI 5.66 to 30.34) in studies recurring to OROV isolation to 0.80% (95%CI 0.31 to 2.06) when all individuals with documented febrile illness in the previous days were sampled with RT-qPCR and 0.03% (95%CI 0.01 to 0.16) when RT-qPCR was the confirmatory test for IFAT. At the same time, the highest detection rate was documented in a single study that relied on RT-qPCR as the confirmatory test of a previous microneutralization test (24.21%, 95%CI 16.65 to 33.81). Similarly, serology was affected by high heterogeneity, with detection rates for IgM ranging from 6.61% (95%CI 0.95 to 34.27) when ELISA was performed as the confirmatory test of a hemagglutination test to 34.83% (95%CI 20.42 to 52.69) in studies without a previous test. Tests such as the hemagglutination test, IFAT, and plaque reduction test, unable to dichotomize between the detected IgG and IgM class antibodies, had corresponding detection rates ranging from 1.03% (95%CI 0.81 to 1.30) for IFAT, to 4.31% (95%CI 1.12 to 15.14) for the plaque reduction test and 24.66% (95% 10.97 to 46.51) for the hemagglutination test. Serology on the general population was again highly heterogenous, with a detection rate for OROV IgG ranging from 29.31% (95%CI 21.91 to 37.99) for ELISA on a hemagglutination test to 59.82% (95%CI 22.71 to 88.29) for ELISA as the only detection test. Finally, we were also able to pool data on CSF samples, with a detection rate of 2.44% (95%CI 1.68 to 5.03) among the sampled cases with a suspected viral infection from high-risk areas.

### 4.2. Generalizability

Having been identified since the mid-1950s [5,7,9,76], OROV can hardly be defined as a “new” virus, being rather an epitome of how neglected tropical infections can suddenly become an emerging infectious disease (EID) of potentially global reach [26,36,37,48]. According to the current WHO definition, neglected tropical diseases (NTDs) are a diverse group of conditions caused by a variety of pathogens (including viruses, bacteria, parasites, fungi, and toxins) and associated with devastating health, social, and economic consequences [94,95,96,97] with several common features. More precisely, NTDs usually affect people living in extreme poverty in sub-Saharan Africa, Asia, Latin America, and the Caribbean, with disproportionately high occurrence in people living below the World Bank poverty figure, producing long-lasting effects either due to chronic infections, chronic disabilities, social stigma, and eventually high morbidity and economic impairment not necessarily translating into high mortality. Until recently, most NTDs were considered somehow “nonemerging” conditions, as they are due to pathogens that have afflicted humanity for centuries, usually in the very same areas where currently documented, envisaging the likelihood of their progressive eradication due to the improvements in the socio-economic status of the affected countries [94,98]. Even though, in the recent decade, large areas of South America and the Caribbean did experience sustained economic growth [99], it came with a large environmental impact and social inequalities [99,100,101] that, in fact, enlarged rather than removed the areas at risk for the spread of arboviruses, including OROV. With about 80% of the total population living in cities [102], not only is South America the most urbanized area in the world, but due to global climate change with heavy rains followed by extreme droughts, uncontrolled deforestation, and the issues associated with the lack of appropriate infrastructures and housing in most urban centers, it has also rapidly become a sort of safe haven for a large number of arthropod species able to host and transmit to human beings a vast array of different pathogens [8]. Not coincidentally, the ongoing OROV epidemics have emerged after the outbreak of ZIKV in 2016 [103,104], while the worst epidemic of Dengue in years is far from being over [105].

The case of OROV infection is of potential global significance for a series of reasons. First, until recently, OROV infections and OF have been considered unable to cause severe consequences, including death and long-lasting disabilities [2,7,31]. However, it should be stressed that our understanding of the true OROV epidemiology has been impaired by the limited diagnostic options and the similarly limited diagnostic opportunities of endemic countries. A key message from our study is that most of the available evidence has been collected on patients with recent history of febrile illnesses. This common blueprint has reasonably resulted in the oversampling of patients who were able to heal from the primary infection, leading to possibly underestimating the fatalities due to OF and OROV infections. In fact, “historical” studies on CSF have documented an otherwise unexpectedly high occurrence rate of neurological complications [71,72], and in post-SARS-CoV-2 pandemic settings, due to the increased availability of high-output diagnostic options (e.g., multiplex RT-qPCR), two fatal cases due to OROV have recently been diagnosed and reported [46]. Moreover, even when dealing with studies of febrile patients, the timing of sample collection and the diagnostic accuracy of the testing option can result in strikingly different estimates for the viral detection rates due to the timing of the OROV viremia. According to our current understanding, OROV viremia peaks on days three and four after the primary infection [106], which is high enough to infect biting midges, and then gradually decreases over the next 31 days, remaining detectable over time. Because of the decrease in viral copies, RT-qPCR on serum is considered reliable only during the first five days of infection [1,7,8]. Therefore, all samples collected well after the viremic peaks could fail to identify underlying viral infection, eventually explaining the very low detection rates in studies that performed RT-qPCR only as a confirmatory test. In effect, studies on the serology of OROV are quite consistent in documenting the high circulation of the pathogen, with prevalence rates that, according to the diagnostic and sampling strategies, could be higher than 50% of the parent population. In other words, our results and the emerging evidence of otherwise unexpected fatal cases suggest that in the past decade, the true disease burden due to OROV may have been extensively underestimated.

Second, while OROV is usually associated with *Culicoides paraensis* midges and *Culex quinquefasciatus* mosquitoes, not only other arthropod species have been shown able to host the pathogen, including *Aedes serratus* and *Coquillettidia venezuelensis*, but also other *Culicoides* species have been proven to be competent vectors [1,5,6,7,27,28]. This is particularly important from a global health perspective, as other zoonotic orthobunyaviruses, such as Schmallenbergvirus (SBV), have been documented in old-world *Culicoides* [107,108]. As SBV is diffused in wild and domestic ruminants across Europe, the limited human reach of SBV infections should be understood as the inability of the pathogen to infect human cells rather than a lack of its circulation [108,109]. Moreover, the biology of orthobunyavirus may contribute to the rapid spread of OROV infections. Due to the segmented nature of the RNA genome, all *Orthobunyaviruses* could possibly reassort in the case of coinfections. In fact, there is some evidence that a large proportion of OF may be due to reassortant viruses rather than to “wild type” OROV strains [70]. This mechanism is believed to have also been involved in the evolution of a human pathogen, Ngari Virus (NRIV), which consists of the L and S segments of Bunyamwera Virus (BUNV) and the M segment of Batai Virus (BATV) [110]. As *Orthobunyaviruses* are endemic to many other areas of the world, including Africa, Europe, Asia, and North America, we cannot rule out that OROV may possibly spread outside Latin America due to the availability of suitable vectors and/or the acquisition of affinity to local vectors following reassorting. Interestingly enough, recent data have documented anti-OROV antibodies even in cattle and dogs from Brazilian urban areas, stressing the extensive zoonotic potential of this pathogen due to the possible reservoir population among common domestic animals [83,91]. The documented similarities between OROV and other *Orthobunyaviruses* represent another significant source of concern. For instance, Akabane Virus, another member of the Simbu serogroup with *Culicoides* midges as documented vectors, is a well-documented teratogen that causes severe fetal damage among domestic animals, particularly in cattle, buffalo, sheep, goats, and even horses [111,112,113]. Fetal defects associated with Akabane Virus infection include extensive CNS involvement (i.e., porencephaly and hydranencephaly) [111,113], and while very little is known about the teratogenic potential of OROV, its tropism for brain tissue and its capability to cross the placental barrier and infect the fetus represent a warning for potential risks faced by pregnant women during OROV outbreaks [45,72,114].

Nonetheless, our study suggests that OROV, despite its potential burden and its significance in terms of global health, appears as significantly less efficient than other arboviruses, such as ZIKV, CHKV, and DENV, in terms of human spread. Even though the cocirculation of these pathogens has been repetitively documented [6,44,64,73,86,115], pooled data suggest that even when and where OROV outbreaks have been reported, viral detection rates for the aforementioned arboviruses may exceed those for OROV several times (i.e., RR 2.207 (95%CI 1.427 to 3.412), RR 7.900 (95%CI 5.386 to 11.578), and RR 24.816 (95%CI 21.119 to 29.159) for the viral detection rates of DENV, ZIKV, and CHKV compared with those of OROV, respectively; RR 3.103 (95%CI 2.056 to 4.865), RR 49.500 (95%CI 12.256 to 199.921), and RR 23.631 (95%CI 20.584 to 27.129) for serology of CHKV, ZIKV, and DENV, respectively). Some explanations may be provided by the underlying biology of their competent vectors. During the four-decade time period included in the present study, Latin America was affected by the extensive re-infestation of *Aedes aegypti*, which resulted in the ongoing major outbreaks of DENV and contributed to a lesser extent to the previous outbreaks of CHKV and ZIKV [116,117,118]. *A aegypti* is well adapted to breeding in close association with humans [117,118] but is not considered a competent vector for OROV, which, in turn, is more frequently associated with arthropods like *Culicoides paraensis* [29,115]. As the *C paraensis* species thrives in close association with agricultural wastes (e.g., husks and banana stalks), OROV can unsurprisingly result in large numbers of human infections within farming communities [1,2,3,17]. On the contrary, its diffusion in the urban environments relies on far less effective vectors, such as *Culex quinquefasciatus*, *Aedes serratus*, *Coquillettidia venezuelensis*, and other *Culicoides* species, ultimately appearing marginal compared with the pathogens otherwise hosted by *A aegypti* (i.e., CHKV, ZIKV, and DENV) [1,2,3].

Unfortunately, far from diminishing the actual burden of OROV infections, the other side of the coin is that arboviruses emerge as a cumulative threat, urging for a more comprehensive approach addressing these pathogens, their hosts, and vectors as a whole [119,120,121]. This One Health approach has been repetitively advocated by global [121,122], regional [123], national, and even local health authorities, but way more rarely implemented [120,121,124,125,126]. In this regard, a success story can be identified in the Italian National Plan for Prevention, Surveillance, and Response to Arbovirosis, which, since 2018, has tracked the circulation of zoonotic viruses such as West Nile Virus, Tick-Borne Encephalitis Virus, Usutu Virus, Toscana Virus, and even DENV, ZIKV, and CHKV [120,121,127].

This integrated surveillance is coordinated by Istituto Superiore di Sanità (ISS) and (for the West Nile and Usutu Viruses) Istituto Zooprofilattico of Abruzzo and Molise (Izs-AM), in collaboration with the Ministry of Health [127,128]. Through the active integration and collaboration of professionals with a background in human medicine, veterinary medicine, entomology, and more recently, even climatology, this integrated surveillance not only regularly publishes surveillance reports but provides the blueprints for specifically tailored response plans to ensure the early detection of potential cases and minimize any spread of diseases [124,125,126,129].

### 4.3. Limitations

Despite its potential significance from a public health point of view, our study is affected by several shortcomings, including the intrinsic limits of systematic reviews and meta-analyses and those associated with the studies we gathered.

First and foremost, the quality of the evidence we reported is dependent on that of the parent studies. In fact, the preventive ROB analysis documented the uneven quality of the source studies, which in turn depends on the heterogeneous sampling and reporting strategies. On the one hand, studies only including subjects affected by febrile illness are limitedly representative when dealing with the actual incidence of OROV infection, as the included series did not take into account asymptomatic and/or afebrile patients. The fact that most early studies reported on outbreaks where OROV was almost certainly the causative agent greatly contributes to the selection bias [10,11,12,60,106], especially when the comparison is made with studies that may have been designed in order to document the occurrence of DENV, CHKV, or ZIKV infections, leading to the heterogeneous sampling we documented across the four-decade time period we included in this study [31,41,86,87,89,90]. On the other hand, while studies on the general population may hint at the cumulative occurrence of OROV infections in a certain area, serology data are of limited reliability when dealing with any tentative reconstruction of their incidence. Moreover, the focal nature of OROV outbreaks, which in turn results from the biology of the documented vectors, could significantly skew estimates of prevalence in what was otherwise considered “general populations” [14,15,63,84,87].

Second, as the studies span over four decades, diagnostic strategies have evolved over time in order to cope with the increasing availability of accurate and effective diagnostic options, but also with the constraints of field research studies [2,10,36,39,130]. In fact, most of the studies were performed in areas within the Amazonas, and samples were obtained in limited-resource areas. Moreover, laboratory techniques have evolved only in recent years by guaranteeing the simultaneous assessment of several pathogens at the same time (e.g., by means of multiplex PCR), leading to the potential oversampling of OROV in earlier studies compared with lower estimates in more recent reports [44,65,67,72,76,78,84,85,89,90].

Third, we must stress the potentially limited representativity of the pooled sample. According to available data, South America’s population ranges between 424 and 442 million people [131]. Focusing on the serology of OROV, the pooled sample included a total of 24,470 cases with any signs or symptoms and 5247 cases from the general population, for a total of around 30,000 people; our study only encompasses around 0.007% of the total population of South America. The pooled sample is also scarcely representative of the actual demographics of the parent countries. Brazil, with around 201 million people, has by far the largest population in South America (around 45.48% to 47.41% of the total population), and also, in our sample, the most significant proportion of cases was documented in Brazil (around 61.39%, including both cases from outbreaks and the general population). However, the second largest population group was collected in Peru, which accounts for around 34 million people, which is around 8% of the population of South America but encompassed 34.63% of our pooled sample and 41.11% of subjects sampled from the general population. The over-representativity of some areas is even more evident when focusing on the detection of viral RNA or viral isolation, as Peru encompassed two-thirds of the pooled sample, and Brazil only contributed around 10% of the sample. Finally, even within these large samples, the representativity of the national level is limited. The large majority of samples were retrieved in the areas of the Amazonas, which are reasonably characterized by the highest prevalence of vectors and pathogens but account for a reduced proportion of the total population. In other words, the pooled estimates reasonably overestimated the actual occurrence of OROV; a quite cautious approach in their generalization is, therefore, highly warranted.

Fourth, our estimates were reasonably affected by a substantial small-study effect. In fact, substantial differences in the detection rates for OROV were identified, with crude estimates for viral RNA and viral cultures of 1.64% compared with a pooled estimate of 0.45% (95%CI 0.16 to 1.23), and similar differences were identified for all other biomarkers. Even though these results may appear quite confusing, it should be stressed that the meta-analytical approach was adopted to better cope with potential confounders and sources of bias associated with source studies than otherwise allowed by the simple cumulative summary of individual data [132,133]. Notably, in this specific case, the simple algebraic sum of the prevalence has reasonably determined the substantial underestimation of the actual prevalence rates for OROV. Therefore, while the high heterogeneity of source studies urges for a quite cautious appraisal of our results, the substantial differences between the crude and pooled estimates reasonably stress how a REM meta-analytical approach likely represents the more appropriate way for handling the highly variable landscape of observational studies on the OROV detection rates.

## 5. Conclusions

OROV, the causal agent of OF, an NTD, has recently been characterized as an emergent zoonotic disease, not only in South America but also from a global perspective. Our systematic review with meta-analysis suggests that OROV infections may be largely underestimated due to the sampling and testing strategies, focusing on the assessment of ongoing outbreaks rather than on the general population. Nonetheless, the characteristics of this pathogen, its documented large array of hosts and vectors, and the role of environmental and socioeconomic factors collectively highlight how OROV could fail to obtain a global reach in the near future. In fact, pooled estimates suggest that DENV, CHKV, and ZIKV are far more effective in causing local epidemics, particularly in urban areas than OROV, possibly due to the underlying biology of competent vectors. However, the recent history of West Nile Virus and, most notably, Dengue, their initial spread in North American and Western European countries, and the subsequent spread to most of western Europe and Eurasia suggest that national and international surveillance plans for arboviruses should include OROV and OF in order to rapidly identify the potential spread of this pathogen outside of Latin America. A comprehensive “One Health” approach, through the collaboration of professionals from human, veterinary medicine, entomology, biology, and environmental sciences, could, therefore, guarantee effective prevention and mitigation strategies.

## Figures and Tables

**Figure 1 viruses-16-01498-f001:**
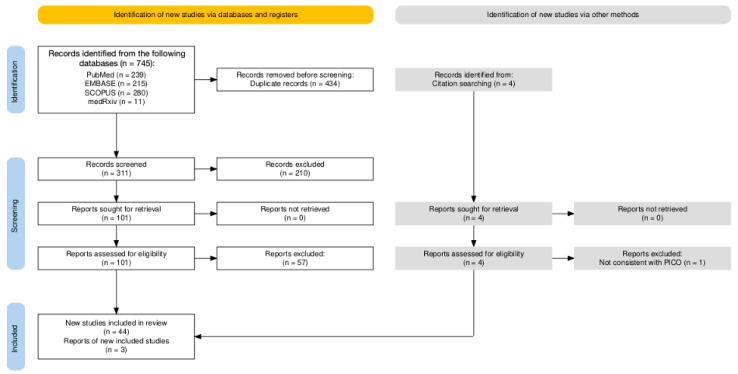
Flow chart of search strategy according to PRISMA 2020 strategy [49,50].

**Figure 2 viruses-16-01498-f002:**
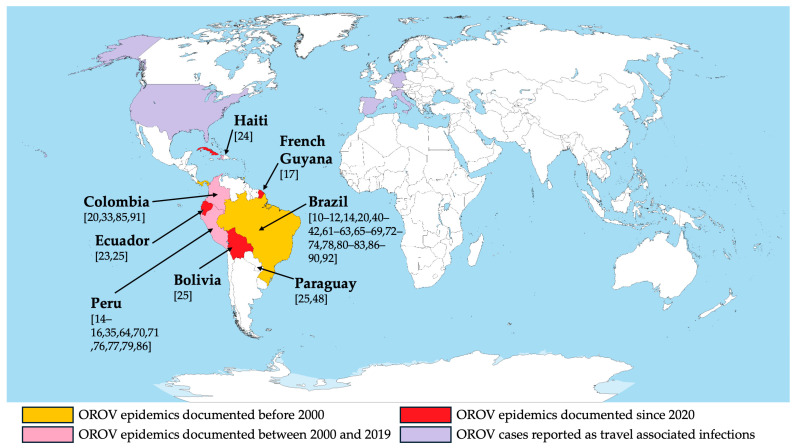
Summary of articles included in a systematic review on the epidemiology of Oropouche Virus infections by geographic location. References to original articles are reported in square brackets [10,11,12,13,15,16,19,22,23,24,32,34,39,40,41,47,60,61,62,63,64,65,66,67,68,69,70,71,72,73,74,75,76,77,78,79,80,81,82,83,84,85,86,87,88,89,90]. (Original file licensed under the GNU Free Documentation License, version 1.2; https://commons.wikimedia.org/wiki/File:A_large_blank_world_map_with_oceans_marked_in_blue.PNG, accessed on 12 September 2024.)

**Figure 3 viruses-16-01498-f003:**
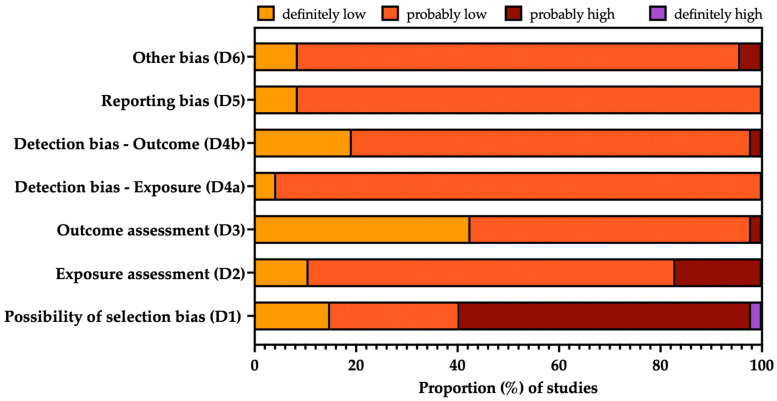
Summary of risk of bias (ROB) analysis of (a) observational studies (performed according to the risk of bias (ROB) tool from the National Toxicology Program (NTP)’s Office of Health Assessment and Translation (OHAT) handbook [54,92]).

**Figure 4 viruses-16-01498-f004:**
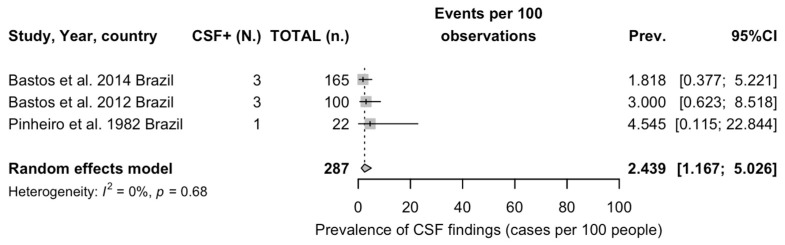
Forest plot on studies reporting on positive status for Oropouche Virus of cerebrospinal fluid (CSF) specimens (note: 95%CI = 95% confidence interval) [60,71,72].

**Table 1 viruses-16-01498-t001:** Summary of articles included in systematic review on epidemiology of Oropouche Virus infections.

Study	Country	Timeframe	Area	Sampling Strategy	Sampled Medium	Laboratory Analysis	Sample Size (N)	Positive (*n*/N, %)
Pinheiro et al., 1976 [10]	Brazil	February 1975–April 1975	State of Pará (Mojoì, Palhal, and nearby villages)	All subjects with acute febrile illness during local outbreak	Blood	Virus isolation	247	69 (27.94%)
						Serology (hemagglutination)	282	119 (42.20%)
LeDuc et al., 1981 [11]	Brazil	July 1978– September 1978 May 1979–June 1979	State of Pará (Quatro Bocas and Tome Acu)	All subjects with acute febrile illness during local outbreak	Blood	Serology (hemagglutination)	555	164 (29.55%)
Pinheiro et al., 1982 [60]	Brazil	1980	State of Pará	All subjects with suspected meningitis/encephalitis	CSF	Virus isolation	22	1 (4.55%)
						Detection of Ig in CSF (hemagglutination)	22	3 (13.64%)
Borborema et al., 1982 [12]	Brazil	May 1981–July 1981	State of Amazonas (Barcelos and Manaus)	All subjects with acute febrile illness during local outbreak	Blood	Serology (hemagglutination)	760	254 (33.42%)
Vasconcelos et al., 1989 [61]	Brazil	1987	State of Maranhao (Porto Franco)	All subjects with acute febrile illness during local outbreak	Blood	Virus isolation	75	22 (29.33%)
						Serology (hemagglutination + ELISA (IgM))	197	128 (64.97%)
Rosa et al., 1996 [62]	Brazil	June 1994	State of Pará	Random sampling among residents	Blood	Virus isolation	296	10 (3.38%)
						Serology (hemagglutination + ELISA (IgG))	296	245 (82.77%)
Watts et al., 1997 [63]	Peru	1992	Loreto Department (Manacamiri, Padre Cocha, Porvenir, Primavera, and Villa Punchana)	Random sampling among residents Follow-up among subjects with negative samples	Blood	ELISA (IgG)	1616	448 (27.72%)
Baisley et al., 1998 [13]	Brazil	June 1996–September 1999	State of Amazonas (Santa Clara)	Random sampling among residents (age > 5 years)	Blood	Serology (hemagglutination + ELISA (IgG))	1227	390 (31.78%)
de [63] Figueredo et al., 2004 [64]	Brazil	March 1998–December 1999	State of Amazonas	All subjects with acute febrile illness during local outbreak	Blood	Serology (hemagglutination + ELISA (IgM))	8557	3 (0.04%)
Tavares-Neto et al., 2004 [65]	Brazil	14 August 1999	State of Acre (Rio Branco)	Random sampling among residents (during HBV vaccination campaign)	Blood	Serology (hemagglutination + ELISA (IgM))	394	9 (2.28%)
da Silva Azevedo et al., 2007 [39]	Brazil	March 2003–May 2003 July 2004–August 2004	State of Pará (Parauapebas and Porto de Moz)	All subjects with acute febrile illness during local outbreak	Blood	Serology (hemagglutination + ELISA (IgM))	234	93 (39.74%)
Bernarders Terzian et al., 2009 [66]	Brazil	March 2004–October 2006	State of Acre	All subjects with acute febrile illness during local outbreak (age > 5 years)	Blood	RT-qPCR	69	1 (1.45%)
						Serology (plaque reduction neutralization test)	357	6 (1.68%)
Cruz et al., 2009 [67]	Brazil	October 2006–December 2007	State of Pará	All subjects with acute febrile illness during local outbreak	Blood	Serology (hemagglutination)	1597	90 (5.64%)
						Serology (hemagglutination + ELISA (IgM))	1597	23 (1.44%)
Manock et al., 2009 [22]	Ecuador	Arril 2001–September 2004	Pastaza Province	All subjects with acute febrile illness during local outbreak	Blood	RT-qPCR	229	0 (-)
						Serology (plaque reduction neutralization test)	304	1 (0.32%)
Mourao et al., 2009 [68]	Brazil	January 2007–November 2008	State of Amazonas (Manaus)	All subjects with acute febrile illness during local outbreak (age > 5 years)	Blood	Serology (ELISA (IgM))	631	128 (20.29%)
Vasconcelos et al., 2009 [40]	Brazil	May 2006–June 2006	State of Pará (Magalhaes Barata and Maracana)	All subjects with acute febrile illness during local outbreak	Blood	Serology (hemagglutination + ELISA (IgM))	744	113 (15.19%)
Alvarez-Falconi et al., 2010 [69]	Peru	May 2010	Loreto Department (Bagazan)	All subjects with acute febrile illness during local outbreak	Blood	Serology (ELISA (IgM))	171	108 (63.16%)
Forshey et al., 2010 [24]	Bolivia	2000–2007	Region of Cochabamba, Conception, Magdalena, and Santa Crus	All subjects with acute febrile illness during local outbreak (age > 5 years) Children with suspected Dengue hemorrhagic fever	Blood	Indirect immunofluorescence assay followed by RT-qPCR	2089	0 (-)
	Ecuador		Region of Guayaquil				350	0 (-)
	Peru		Departments of Iquitos, La Merced, Padre Maldonado, Piura, Tumbes, and Yurimaguas				18201	18 (0.10%)
	Paraguay		Region of Asunción				240	0 (-)
Aguilar et al., 2011 [70]	Peru	1995–2006	Loreto Department (Iquitos)	All subjects with acute febrile illness during local outbreak	Blood	Serology (plaque reduction neutralization test)	1037	154 (14.85%)
		2006					1037	2 (0.19%)
Bastos et al., 2012 [71]	Brazil	2005–2010	Amazonas	All subjects with suspected CNS infection	CSF	RT-qPCR	100	3 (3.00%)
Bastos et al., 2014 [72]	Brazil	January 2010–August 2012	Amazonas	All subjects with suspected CNS infection	CSF	RT-qPCR	165	3 (1.82%)
Martins et al., 2014 [73]	Brazil	January 2011–May 2011	Amazonas (Manaus)	All subjects with acute febrile illness during local outbreak	Blood	RT-qPCR	677	0 (-)
Cardoso et al., 2015 [74]	Brazil	October 2011–July 2012	Mato Groso	All subjects with acute febrile illness during local outbreak	Blood	RT-qPCR	529	5 (0.95%)
Garcia et al., 2016 [75]	Peru	13 December 2014–8 January 2016	Madre de Dios	Specimens from individuals sampled for Dengue and Leptospira	Blood	RT-qPCR	508	19 (3.74%)
						Virus isolation	508	32 (6.30%)
						Serology (ELISA (IgM))	508	122 (24.02%)
Alva-Urcia et al., 2017 [76]	Peru	January 2016–March 2016	Madre de Dios	All subjects with acute febrile illness during local outbreak	Blood	RT-qPCR	139	12 (8.63%)
Silva-Caso et al., 2019 [34]	Peru	January 2016–July 2016	Huanuco Region	Adults with acute febrile illness lasting < 7 days	Blood	RT-qPCR	268	46 (17.16%)
do Nascimiento et al., 2020 [77]	Brazil	February 2016–June 2016	Amazonas	All subjects with acute febrile illness during local outbreak	Blood	RT-qPCR	352	5 (1.42%)
Martins-Luna et al., 2020 [78]	Peru	February 2016–September 2016	Piura Region	Adults with acute febrile illness	Blood	RT-qPCR	496	131 (26.41%)
Rojas et al., 2020 [47]	Paraguay	April 2019	Not reported	Random sampling among people with suspected arboviral illness	Blood	RT-qPCR	100	0 (-)
Salvador et al., 2020 [79]	Brazil	November 2016–December 2017	Bahia (Salvador)	Subjects referring to a private local hospital with Dengue-like symptoms	Blood	RT-qPCR	53	2 (3.77%)
Elbadry et al., 2021 [23]	Haiti	2014	Gressier	All children from a local school with an acute febrile illness	Blood	RT-qPCR	1250	1 (0.08%)
Gaillet et al., 2021 [16]	French Guyana	11 August 2020–15 October 2020	Saúl	All cases with Dengue-like symptoms	Blood	RT-qPCR on microneutralization test	95	23 (24.21%)
Saatkamp et al., 2021 [80]	Brazil	2016	State of Pará	All febrile subjects of adult age with an acute febrile status	Blood	RT-qPCR	49	0 (-)
Salgado et al., 2021 [81]	Brazil	Jaunuary 2014–December 2015	Nationwide	Random sampling from Brazilian armed forces	Blood	Serology (Hemagglutination test)	298	2 (0.67%)
Carvalho et al., 2022 [82]	Brazil	Jaunuary 2018–February 2018	State of Pará	All subjects reporting a febrile illness in the previous 30 days or had contact with them	Blood	Virus isolation	94	14 (14.89%)
						Serology (ELISA (IgM))	94	36 (38.30%)
Ciuoderis et al., 2022 [32]	Colombia	February 2019–Jaunuary 2022	Regions of Calì, Cucuta, Leticia, and Villavicencio	All febrile subjects aged over 5 years	Blood	RT-qPCR	2967	105 (3.54%)
Dias et al., 2022 [83]	Brazil	February 2016–March 2016	Mato Groso	Retrospective analysis of samples collected in subjects with a febrile illness from < 7 days	Blood	RT-qPCR	106	0 (-)
Gil-Mora et al., 2022 [84]	Colombia	2018	Cauca Department	All subjects > 18 years from the parent municipalities	Blood	Serology (plaque reduction neutralization test)	505	10 (1.98%)
Gonçalves Maciel et al., 2022 [85]	Peru	February 2018–May 2019	Puerto Maldonado, Piura, and Huanuco	All subjects reporting skin rash	Blood	RT-qPCR	340	0 (-)
Watts et al., 2022 [15]	Peru	1993–1997	Loreto department	All febrile subjects aged 1 to 60 years, symptoms lasting < 5 days	Blood	IFAT	6607	68 (1.03%)
De Lima et al., 2024 [86]	Brazil	August 2014–May 2015	Amapà region	All subjects with acute febrile illness during local outbreak	Blood	RT-qPCR	166	0 (-)
						Serology (plaque reduction neutralization test)	166	17 (10.24%)
Forato et al., 2024 [41]	Brazil	December 2018–December 2021	Roraima (11 municipalities)	All subjects with acute febrile illness during local outbreak	Blood	RT-qPCR	883	0 (-)
Grisales-Nieto et al., 2024 [19]	Colombia + Brazil	November 2020	Department of Leticia and Amazonas (Colombia) and State of Amazonas (Brazil)	Random sampling from residents negative to other arboviruses	Blood	RT-qPCR	175	1 (0.57%)
Moreira et al., 2024 [87]	Brazil	January 2022–March 2023	States of Rondonia and Amazonas	All subjects with acute febrile illness lasting 5 to 7 days during local outbreak Excluded indigenous people, pregnant women, and all subjects with positive testing for other arboviruses	Blood	RT-qPCR	351	27 (7.69%)
Scachetti et al., 2024 [88]	Brazil	December 2023–March 2024	State of Amazonas	All subjects with acute febrile illness during local outbreak	Blood	RT-qPCR	93	10 (10.75%)
Silva et al., 2024 [89]	Brazil	2019–2020	State of Amazonas	All subjects > 3 years from the communities of Cararà and Espirito Santo	Blood	Serology (immunochromatography IgM)	205	0 (-)
						Serology (immunochromatography IgG)	205	0 (-)
Sanchez-Lerma et al., 2024 [90]	Colombia	January 2021–June 2023	Department of Meta	All subjects with acute febrile illness during local outbreak	Blood	RT-qPCR	100	0 (-)

Note: RT-qPCR = real-time polymerase chain reaction; IFAT = indirect immunofluorescent antibody test; CNS = central nervous system; CSF = cerebrospinal fluid.

**Table 2 viruses-16-01498-t002:** Occurrence of Oropouche Virus (i.e., detection of either virus isolates or RNA) from samples collected from subjects with any sign/symptom of arbovirus infection.

	Total Samples (N)	OROV-Positive Cases (*n*/N, %)	Risk Ratio (95%CI)
Overall	32,439	532, 1.64%	-
Timeframe			
Before 1990	344	92, 26.74%	REFERENCE
1991–2010	21,178	19, 0.09%	0.003 (0.002; 0.005)
After 2010	10,917	421, 3.86%	0.144 (0.118; 0.175)
Country			
Bolivia	2089	0, -	0.006 (0.001; 0.096)
Brazil	4030	162, 4.02%	REFERENCE
Colombia	3067	105, 3.42%	0.852 (0.669; 1.084)
Ecuador	579	0, -	0.021 (0.001; 0.344)
French Guyana	95	23, 24.21%	6.023 (4.092; 8.864)
Haiti	1250	1, 0.08%	0.020 (0.003; 0.142)
Paraguay	340	0, -	0.037 (0.002; 0.586)
Peru	20,989	241, 1.15%	0.286 (0.235; 0.348)
Sample			
Blood	32,152	525, 1.63%	REFERENCE
CSF	287	7, 2.44%	1.494 (0.715; 3.120)
Diagnostic procedure			
Virus isolation	946	138, 14.59%	4.347 (3.612; 5.231)
RT-qPCR	10,518	353, 3.36%	REFERENCE
RT-qPCR + IFAT	20,880	18, 0.08%	0.026 (0.016; 0.041)
RT-qPCR + MNT	95	23, 24.21%	7.214 (4.981; 10.446)

Note: OROV = Oropouche Virus; 95%CI = 95% confidence interval; RT-qPCR = real-time polymerase chain reaction; CSF = cerebrospinal fluid; IFAT = indirect immunofluorescence assay; MNT = microneutralization test.

**Table 3 viruses-16-01498-t003:** Seroprevalence studies on antibodies targeting the Oropouche Virus from samples collected from subjects with any sign/symptom of arbovirus infection.

	Total Samples (N)	OROV-Positive Cases (*n*/N, %)	Risk Ratio (95%CI)
Overall	24,470	1630, 6.66%	-
Timeframe			
Before 1990	1816	668, 36.78%	REFERENCE
1991–2010	14,192	565, 3.98%	0.108 (0.097; 0.120)
After 2010	8462	397, 4.69%	0.128 (0.114; 0.143)
Country			
Bolivia	2089	0, -	0.003 (0.001; 0.051)
Brazil	15,793	1177, 7.45%	REFERENCE
Colombia	50	0, -	0.134 (0.009; 2.117)
Ecuador	304	1, 0.33%	0.044 (0.006; 0.313)
Peru	8323	452, 5.43%	0.729 (0.656; 0.809)
Antibody			
IgG/IgM	13,030	898, 6.89%	1.077 (0.980; 1.184)
IgM	11,440	732, 6.40%	REFERENCE
Sample			
Blood	24,448	1627, 6.65%	REFERENCE
CSF	22	3, 13.64%	2.049 (0.715; 5.871)
Diagnostic procedure			
ELISA	13,037	755, 5.79%	REFERENCE
HAT	3216	630, 19.59%	3.383 (3.065; 3.733)
PRT	1610	177, 10.99%	1.898 (1.625; 2.217)
IFAT	6607	68, 1.03%	0.178 (0.139; 0.227)

Note: OROV = Oropouche Virus; 95%CI = 95% confidence interval; ELISA = enzyme-linked immune assay; HAT = hemagglutination test; CSF = cerebrospinal fluid; IFAT = indirect immunofluorescence assay; PRT = plaque reduction neutralization test.

**Table 4 viruses-16-01498-t004:** Seroprevalence studies on antibodies targeting the Oropouche Virus from samples collected from the general population.

	Total Sample (N)	OROV-Positive Cases (*n*/N, %)	Risk Ratio (95%CI)
Overall	5247	1113, 21.21%	-
Timeframe			
Before 1990	0	-	-
1991–2010	4085	1101, 26.95%	REFERENCE
After 2010	1162	12, 1.03%	0.038 (0.022; 0.067)
Country			
Brazil	2625	646, 24.61%	REFERENCE
Colombia	455	10, 2.20%	0.089 (0.048; 0.165)
Peru	2168	457, 21.08%	0.857 (0.771; 0.952)
Antibody			
IgG	3896	1092, 28.03%	18.655 (9.735; 35.746)
IgG/IgM	753	12, 1.59%	1.061 (0.450; 2.500)
IgM	599	9, 1.50%	REFERENCE
Diagnostic procedure			
ELISA	4085	1101, 26.95%	REFERENCE
HAT	298	2, 0.67%	0.025 (0.006; 0.099)
PRT	455	10, 2.20%	0.082 (0.044; 0.151)
ICT	410	0, -	0.005 (0.001; 0.072)

Note: OROV = Oropouche Virus; 95%CI = 95% confidence interval; ELISA = enzyme-linked immune assay; HAT = hemagglutination test; PRT = plaque reduction neutralization test; ICT = immunochromatographic test.

**Table 5 viruses-16-01498-t005:** Occurrence of arboviral infections, only paired studies.

Pathogen	Tested (N)	OROV Positive (*n*/N, %)	Arbovirus Positive (*n*/N, %)	Risk Ratio (95%CI)
Virology				
DENV	27,660	152, 0.55%	3772, 13.64%	24.816 (21.119; 29.159)
ZIKV	2758	29, 1.05%	229, 8.30%	7.900 (5.386; 11.578)
CHKV	2924	29, 0.99%	64, 2.19%	2.207 (1.427; 3.412)
Serology				
DENV	20,269	206, 1.02%	4868, 24.02%	23.631 (20.584; 27.129)
ZIKV	708	2, 0.28%	99, 13.98%	49.500 (12.256; 199.921)
CHKV	1379	29, 2.10%	90, 6.53%	3.103 (2.056; 4.685)

Note: OROV = Oropouche Virus; 95%CI = 95% confidence interval; DENV = Dengue Virus; ZIKV = Zika Virus; CHKV = Chikungunya Virus.

**Table 6 viruses-16-01498-t006:** Meta-analysis of the prevalence of Oropouche Virus isolates among subjects with signs/symptoms of arbovirus infection; blood specimens (note: 95%CI = 95% confidence interval).

Testing	No. of Series	No. of Observations	No. of Events	Detection Rate (%, 95%CI)	I^2^ (95%CI)	Tau^2^	Q	*p*-Value
Virological assay								
Overall	49	33,156	553	0.45 (0.16 to 1.23)	94.9% (85.7 to 93.9)	9.647	942.13	<0.001
Virus isolation	8	1420	146	13.91 (5.66 to 30.34)	95.3%	1.875	147.87	
RT-qPCR	27	10,761	18	0.80 (0.31 to 2.06)	93.5%	5.585	402.38	
RT-qPCR (on IFAT)	13	20,880	366	0.03 (0.01 to 0.16)	0.0%	1.115	0.359	
RT-qPCR (on MNT)	1	95	23	24.21 (16.65 to 33.81)	-	-	-	
Serology, IgM only								
Overall	12	13,037	755	12.45 (3.28 to 37.39)	98.5% (98.1 to 98.8)	6.310	726.71	<0.001
ELISA	4	1404	394	34.83 (20.42 to 52.69)	97.4%	0.539	114.57	
ELISA (on HAT)	8	11633	361	6.61 (0.95 to 34.27)	98.8%	8.124	596.32	
Serology, IgM/IgG								
Overall	14	11,411	872	12.21 (4.96 to 27.09)	99.0% (98.8 to 99.2)	3.318	1337.11	<0.001
HAT	9	3194	627	24.66 (10.97 to 46.51)	98.7%	2.143	629.52	
PRT	4	1610	177	4.31 (1.12 to 15.14)	90.6%	1.501	31.90	
IFAT	1	6607	68	1.03 (0.81 to 1.30)	-	-		

Note: I^2^ = percent proportion of variance in study estimates that is due to heterogeneity; tau^2^ = estimate of variance of underlying distribution of true effect sizes; Q = weighted sum of squared differences between individual study effects and pooled effect across studies; 95%CI = 95% confidence interval; RT-qPCR = real-time quantitative polymerase chain reaction; IFAT = indirect immunofluorescence analysis; MNT = microneutralization test; ELISA = enzyme-linked immune assay; HAT = hemagglutination test; PRT = plaque reduction neutralization test.

**Table 7 viruses-16-01498-t007:** Meta-analysis of the prevalence of Oropouche Virus isolates among subjects from the general population (i.e., without documented signs/symptoms of ongoing arbovirus infection).

Testing	No. of Series	No. of Observations	No. of Events	Detection Rate (%, 95%CI)	I^2^ (95%CI)	Tau^2^	Q	*p*-Value
Serology, IgG								
Overall	8	3344	1083	24.45 (7.83 to 55.21)	97.6% (96.5 to 98.3)	3.548	287.63	<0.001
ELISA	2	1616	338	59.82 (22.71 to 88.29)	99.5%	1.357	198.31	
ELISA (on HAT)	5	1523	635	29.31 (21.91 to 37.99)	92.6%	0.176	54.26	
ICT	1	205	0	0.0 (0.00 to 100)	-	-	-	
Serology, IgM/IgG								
Overall	4	753	12	1.42 (0.61 to 3.28)	41.9% (0.0 to 80.5)	0.251	5.17	0.160
HAT	1			0.67 (0.17 to 2.64)	-	-	-	
PRT	3			2.16 (0.99 to 4.68)	6.2%	0.018	2.13	

Note: I^2^ = percent proportion of variance in study estimates that is due to heterogeneity; tau^2^ = estimate of variance of underlying distribution of true effect sizes; Q = weighted sum of squared differences between individual study effects and pooled effect across studies; 95%CI = 95% confidence interval; RT-qPCR = real-time quantitative polymerase chain reaction; ICT = immunochromatography; ELISA = enzyme-linked immune assay; HAT = hemagglutination test; PRT = plaque reduction neutralization test.

**Table 8 viruses-16-01498-t008:** Summary of results of Egger’s test performed on main findings reported in this meta-analysis.

Finding	t	df	Bias (SE)	Tau^2^	*p*-Value
Symptomatic individuals					
Virological assay	−4.37	47	−3.745 (0.857)	16.964	<0.001
Serology, IgM only	−0.84	10	−4.607 (5.466)	67.851	0.419
Serology, IgM + IgG	0.20	12	1.137 (5.796)	111.413	0.848
CSF findings	1.03	1	1.587 (1.543)	0.377	0.491
General population					
Serology, IgG only	0.16	6	0.817 (5.091)	50.087	0.878
Serology, IgM + IgG	−1.42	2	−1.871 (1.318)	1.287	0.292

Note: df = degree of freedom; SE = standard error; CSF = cerebrospinal fluid.

## Data Availability

Source data are available upon request to the corresponding authors.

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
