# Peer review of "(Re-)Emergence of Oropouche Virus (OROV) Infections: Systematic Review and Meta-Analysis of Observational Studies"

_viruses, 2024, doi:10.3390/v16091498_

Round 1

Reviewer 1 Report

Comments and Suggestions for Authors

This manuscript presents an extensive review and metanalysis of publications dealing with Oropouche virus (OROV). 

The title of the manuscript is: “Epidemiology and clinical features of Oropouche virus (OROV): systematic review and metanalysis;” however, the authors fail to adequately address significant aspects of the epidemiology of OROV that impact their analysis.

The study is challenged by attempting to systematically review all publications over more than a four-decade timeline.  During this period, virological techniques evolved considerably and epidemiological conditions changed dramatically, especially with the re-infestation of Latin America by Aedes aegypti and the resulting major outbreaks of dengue, and to a lesser extent Zika and chikungunya viruses.

Early studies of OROV outbreaks that occurred before about 1990 offer a clearer assessment of the epidemiology of ORO virus and clearly established the importance of Culicoides paraenesis as the primary vector (See Pinheiro et al, 1982, Science 215:1251-1253).  The biology of C. paraenesis is quite different from Ae. aegypti and is most often associated with banana or cacao farming where the Culicoides breed in discarded husks and banana stalks.   This leads to outbreaks that may result in large numbers of human infections but that may be limited in scope to areas where the C. paraenesis are abundant.  This is unlike dengue and other viruses that are transmitted by Ae. aegypti that is well adapted to breeding in close association with humans in general.  This fundamental difference suggests that the risk of global spread of OROV may not be as great as suggested by the authors.

The fact that early studies reported on outbreaks where OROV was almost certainly the causative agent contributes greatly to the selection bias, especially when comparison is made to later studies that may have been in response to dengue or other viruses.  Similarly, the focal nature of OROV outbreaks could significantly skew estimates of prevalence in what is referred to as “general populations.”

The authors describe important observations regarding the occurrence of OROV in CNS.  Historically, CNS involvement has been reported among OROV patients, but this study helps to quantify the risks of such complications.

While not mentioned in this manuscript, Akabani virus is another Culicoides-transmitted virus related to OROV that is known to cause serious fetal damage among domestic animals.  As we gain a better understanding of the clinical aspects of OROV infection, we should be alert to the potential risks to pregnant women.

Other suggestions:

Table 6.  Meta-analysis of the prevalence of OROV isolates.  It would be helpful to more clearly explain the column headings and how to interpret them for those readers less familiar with this manner of assessment.

Table 7.  It is not clear what is related in this table.  “Meta-analysis of the prevalence of Oropouche virus isolates among subjects without signs/symptoms of arbovirus infection.”   Why would virus isolation attempts be made on patients without signs/symptoms?  The “detection rate”  and “testing” columns have only serological tests.

Typo on line 568

Lines 662-667.   “..large number of arthropod species able to host and spread…including OROV”  and “all of them share the common feature of being associated with thriving populations of competent arthropod vectors.”    These statements may be speculative and not accurately reflect the risks of OROV transmission.

Lines 712-714.  Some might consider this statement to be a bit alarmist and speculative.

Lines 805-806.  “..hint at the extensive cocirculation of OROV with other arboviruses such as dengue…”  See comments about on epidemiology of OROV as to why extensive cocirculation may be unlikely.

Lines 808-810.  “..OROV could obtain a global reach quite rapidly in the near future.”   Speculative and not justified by what we know about the epidemiology of OROV.

Lines 813-814.  Good point to include OROV in surveillance plans.

Author Response

Estimated Reviewer,

first of all, thank you so much for your collaborative comments, whose content has reasonably addressed some shortcomings of our study, and whose fulfilment will improve radically our study.

More precisely:

1) The title of the manuscript is: “Epidemiology and clinical features of Oropouche virus (OROV): systematic review and metanalysis;” however, the authors fail to adequately address significant aspects of the epidemiology of OROV that impact their analysis.

REPLY: Thank you for your comment; in fact the title is not well correlated with the content of the study, and was therefore edited as follows: "(Re)Emergence of Oropouche Virus (OROV) infections: systematic review and metanalysis of observational studies".

2) The study is challenged by attempting to systematically review all publications over more than a four-decade timeline.  During this period, virological techniques evolved considerably and epidemiological conditions changed dramatically, especially with the re-infestation of Latin America by Aedes aegypti and the resulting major outbreaks of dengue, and to a lesser extent Zika and chikungunya viruses [...].

REPLY: We are deeply in debt with the present reviewer not only for this comment, but also for its very content. We have amended the main text in order to include as much as possible the informations provided within this comments and the subsequent lines. More precisely:

a) OROV has no known interhuman spreading: the virus is transmitted to humans through the bite of infected insects, including the Culicoides paraensis midges (predominant in urban settings) and Culex quinquefasciatus mosquito, but other arthropod species have been documented as harboring the pathogen (e.g. Aedes serratus, Coquillettidia venezuelensis, other Culicoides species) [1,5–7,28,29]. While the importance of Culicoides paraensis has been documented since the early study of Pinheiro et al. in 1982 [30], the primary vertebrate host of OROV, if any, has not been reported. 

b) (from discussion) Some explanations may be provided by the underlying biology of their competent vectors. During the four decades time period included in the present study, Latin America was affected by the extensive re-infestation of Aedes aegypti that resulted in the ongoing major outbreaks DENV and contributed to a lesser extent in the previous outbreaks of CHKV and ZIKV [116–118]. A aegypti is well adapted to breeding in close association with humans [117,118], but is not considered a competent vector for OROV that, in turn, is more frequently associated with arthropods like Culicoides paraensis [30,115]. As the C paraensis species thrives in close association with agricultural wastes (e.g. husks and banana stalks), OROV can unsurprisingly result in large numbers of human infections within farming communities [1–3,18]. On the contrary, its diffusion in the urban environments relies on far less effective vectors such as Culex quinquefasciatus, Aedes serratus, Coquillettidia venezuelensis, and other Culicoides species, ultimately appearing as marginal compared to the pathogens otherwise hosted by A aegypti (i.e. CHKV, ZIKV, and DENV) [1–3].

c) (discussion, limits) In fact, most of the studies were performed from areas within the Amazonas, and samples were obtained in limited resource areas. Moreover, only in recent years laboratory techniques have been evolved by guaranteeing the simultaneous assessment of several pathogens at the same time (e.g. by means of multiplex PCR), leading to the potential oversampling of OROV from earlier studies compared to lower estimates from more recent reports [45,66,68,73,77,79,85,86,90,91].

3) The fact that early studies reported on outbreaks where OROV was almost certainly the causative agent contributes greatly to the selection bias, especially when comparison is made to later studies that may have been in response to dengue or other viruses.  Similarly, the focal nature of OROV outbreaks could significantly skew estimates of prevalence in what is referred to as “general populations.”

REPLY: we amended the text as follows:

a) The fact that most early studies reported on outbreaks where OROV was almost certainly the causative agent greatly contributes to the selection bias [10–12,61,106], especially when the comparison is made to studies that may have been designed in order to document the occurrence of DENV, CHKV, or ZIKV infections, leading to the heterogenous sampling we documented across the four decades time period we included in this study [32,42,87,88,90,91]. On the other hand, while studies on the general population may hint at the cumulative occurrence of OROV infections in a certain area, serology data are of limited reliability when dealing with any tentative reconstruction of their incidence. Moreover, the focal nature of OROV outbreaks, which in turn results from the biology of documented vectors, could significantly skew estimates of prevalence in what was otherwise considered as “general populations” [15,16,64,85,88].

4) While not mentioned in this manuscript, Akabani virus is another Culicoides-transmitted virus related to OROV that is known to cause serious fetal damage among domestic animals.  As we gain a better understanding of the clinical aspects of OROV infection, we should be alert to the potential risks to pregnant women.

REPLY: thank you so much for this comment; in fact, while gathering available evidence on Akabanevirus several similarities between OROV and this pathogen did surface, stressing the importance of this issue. Therefore, discussion was implemented as follows:

The documented similarities between OROV and other Orthobunyaviruses represent another significant source of concern. For instance, Akabane virus, another member of the Simbu serogroup with a Culicoides midge as documented vector, is a well-documented teratogen that causes severe fetal damage among domestic animals, particularly in cattle, buffalo, sheep, goat, and even horses [111–113]. Fetal defects associated with Akabane virus infections include an extensive CNS involvement (i.e. porencephaly, hydranencephaly) [111,113], and while very little is known about the teratogenic potential of OROV, its tropism for brain tissue, and its capability to cross the placental barrier and infect the fetus represent a certain warning for potential risks faced by pregnant women during OROV outbreaks [46,73,114].

5) Table 6.  Meta-analysis of the prevalence of OROV isolates.  It would be helpful to more clearly explain the column headings and how to interpret them for those readers less familiar with this manner of assessment.

REPLY: thank you for your comment; the captions of the tables including meta-analysis data were amended as follows:

Note: I2 = percent proportion of the variance in study estimates that is due to heterogeneity; tau2 = estimate of the variance of the underlying distribution of true effect sizes; Q = weighted sum of squared differences between individual study effects and the pooled effect across studies; 95%CI = 95% confidence interval; RT-qPCR = real-time quantitative polymerase chain reaction; IFAT = indirect immunofluorescence analysis; MNT = microneutralization test; ELISA = enzyme-linked immune assay; HAT = hemagglutination test; PRT = plaque reduction neutralization test.

6) Table 7.  It is not clear what is related in this table.  “Meta-analysis of the prevalence of Oropouche virus isolates among subjects without signs/symptoms of arbovirus infection.”   Why would virus isolation attempts be made on patients without signs/symptoms?  The “detection rate”  and “testing” columns have only serological tests.

REPLY: we amended the labels and captions as follows: 

Table 7. Meta-analysis of the prevalence of Oropouche virus isolates, among subjects from the general population (i.e. without documented signs/symptoms of ongoing arbovirus infection).

Note: I2 = percent proportion of the variance in study estimates that is due to heterogeneity; tau2 = estimate of the variance of the underlying distribution of true effect sizes; Q = weighted sum of squared differences between individual study effects and the pooled effect across studies; 95%CI = 95% confidence interval; RT-qPCR = real-time quantitative polymerase chain reaction; ICT = immunochromatography; ELISA = enzyme-linked immune assay; HAT = hemagglutination test; PRT = plaque reduction neutralization test.

7) Typo on line 568

REPLY: it was fixed, thank you.

8) Lines 662-667.   “..large number of arthropod species able to host and spread…including OROV”  and “all of them share the common feature of being associated with thriving populations of competent arthropod vectors.”    These statements may be speculative and not accurately reflect the risks of OROV transmission.

REPLY: following the implementation of the evidence you suggested, we also agreed with your comments and the referred lined were amended accordingly:

South America is not only the most urbanized area in the world , but the global climate change with heavy rains followed by extreme droughts, uncontrolled deforestation, and the issues associated with the lack of appropriate infrastructures and housing in most urban centers, it has rapidly become a sort of safe haven for a large number of arthropod species able to host and spread to the human beings a vast array of different pathogens [8]. Not coincidentally, the ongoing OROV epidemics have emerged after the outbreak of ZIKV in 2016 [103,104], while the worst epidemic of dengue in years is far from being over [105].

9) Lines 712-714.  Some might consider this statement to be a bit alarmist and speculative.

REPLY: again, as for the previous comment, we revised the main text according to the evidence you provided and the main text was amended and downscaled:

As Orthobunyaviruses are endemic to many other areas of the world, including Africa, Europe, Asia, and North America, we cannot rule out that OROV may possibly diffuse out of Latin America due to the availability of suitable vectors and/or the acquisition of affinity to local vectors following reassorting

10) Lines 805-806.  “..hint at the extensive cocirculation of OROV with other arboviruses such as dengue…”  See comments about on epidemiology of OROV as to why extensive cocirculation may be unlikely.

REPLY: again, as above (we removed extensive and revised the following lines in order to provide the limits for a global scope of the current OROV outbreaks):

Even though the cocirculation of these pathogens has been repetitively documented [6,45,65,74,87,115], 

11) Lines 808-810.  “..OROV could obtain a global reach quite rapidly in the near future.”   Speculative and not justified by what we know about the epidemiology of OROV.

REPLY: we totally agreed with your comments, and we revised the main text as follows:

Nonetheless, the characteristics of this pathogen, its documented large array of hosts and vectors, and the role of environmental and socioeconomic factors, collectively highlight how OROV could fail to obtain a global reach in the near future. In fact, pooled estimates suggest that DENV, CHKV, and ZIKV are far more effective in causing local epidemics, particularly in urban areas, than OROV, possibly due to the underlying biology of competent vectors.

12) Lines 813-814.  Good point to include OROV in surveillance plans.

REPLY: thank you. However, we downgraded the statement in order to be in tone with the revised text, as follows:

However, the recent history of West Nile Virus, and most notably Dengue, their initial involvement of North American and West European countries, and the subsequent spreading to most of Western Europe and Eurasia, suggest that national and international surveillance plans for arboviruses should include OROV and OF in order to rapidly identify the potential spread of this pathogen outside of Latin America. 

Moreover, we would stress that (according to the requests of REv.2) we implemented a Map of south America including all the references to the gathered studies. Similarly, we amended the main text by explaining why table 7 does not include meta-analysis on viral isolation data:

As only two studies documented viral isolates, by means of either RT-qPCR (estimated detection rate for OROV, 0.57%) [20], and virus isolation (estimated detection rate: 3.38%) [63], a meta-analysis of corresponding pooled estimates was not performed.

Finally, we are confident that the revised text may be perceived as appropriate for publication on Viruses, and we again thank you for the comments and the suggestions you provided.

Reviewer 2 Report

Comments and Suggestions for Authors

Oropuche virus (OROV) is an emergent zoonotic virus, that mainly impacts South America, but with increasing potential to affect human population globally. In this systematic review with meta-analysis on the epidemiology of OROV and its disease (OF), the authors analyzed a total of 47 observational studies spanning from 1975 to 2024. The study is well detailed in methods and overall well-written.

This reviewer does not have major comments. The review suffers from a heavy focus on a very detailed Table, what is great! However, as a minor point, the authors may consider including a world map with OROV infections reported in the late 1970s compared to 2023/24 reports, based on the findings they report.

Author Response

Estimated Reviewer,

to begin with, thank you for your collaborative and very positive comments.

Regarding your suggestions, i.e. "However, as a minor point, the authors may consider including a world map with OROV infections reported in the late 1970s compared to 2023/24 reports, based on the findings they report", we've implemented a new figure (Figure 1) that reports all of the included studies with the area where the samples were collected from.

We hope that this implementation may have increased the readability of the full paper.

On the behalf of all Authors, thank you again

MR